# Symptomatic Bunionette Treated with Sliding Oblique Metatarsal Osteotomy—Case Series

**DOI:** 10.3390/jcm11143962

**Published:** 2022-07-07

**Authors:** Yu-Ting Shen, Peng-Ju Huang, Shu-Jung Chen, Shun-Min Chang

**Affiliations:** 1Department of Orthopaedics, Kaohsiung Medical University Hospital, No 100 Tzu-You 1st Road, Kaohsiung City 807, Taiwan; kobe241220@gmail.com (Y.-T.S.); susu1221@gmail.com (S.-J.C.); 2College of Medicine, Kaohsiung Medical University, No 100 Tzu-You 1st Road, Kaohsiung City 807, Taiwan; 3Department of Orthopaedics, Kaohsiung Municipal Ta-Tung Hospital, No. 68, Jhonghua 3rd Road, Kaohsiung City 801, Taiwan; dtorth758@gmail.com

**Keywords:** bunionette, sliding oblique metatarsal osteotomy

## Abstract

Background: The purpose of this study is to investigate the clinical and radiological results of a sliding oblique metatarsal osteotomy (SOMO) to correct bunionette deformity. Methods: We retrospectively reviewed 44 patients (51 feet, left/right: 29/22) from December 2010 to December 2018 who underwent SOMO and compared radiographic measurements and clinical outcome scores preoperatively and postoperatively. Radiographic measurements included 4th and 5th intermetatarsal angle (IMA), metatarsophalangeal angle (MTPA), and lateral deviation angle (LDA). Clinical outcome measurements included The American Orthopedic Foot and Ankle Society (AOFAS) score for lesser metatarsophalangeal procedures and visual analog scale (VAS) pain score. The mean follow-up period was 26.6 months (minimum 18 months). Based on Coughlin and Fallat classification, all cases were separated into four subtypes: 6 type I, 10 type II, 12 type III, 23 type IV cases included.) Results: All radiographic parameters significantly improved after SOMO procedure (IMA/MTPA/LDA, *p* value < 0.001). Clinical scores also showed a significant improvement in AOFAS and VAS scores (*p* value < 0.001). In terms of subgroup based on each type, both radiographic measurements and clinical scores revealed significant improvements in each subgroup (*p* value < 0.05), except LDA of type I subgroup (*p* value = 0.09). Three cases reported pin-tract infection but recovered with good healing after removal of the K-wire and a prescription of oral antibiotic. Conclusion: The SOMO procedure may be considered as a reliable and simple treatment for most types of bunionette deformity with satisfactory outcomes and no severe complications. Level of Evidence: Level IV, case series.

## 1. Introduction

Bunionette, which is also called Tailor’s bunion, is a common disease of the fifth toe. The bony anomalies of bunionette on the fifth metatarsal that can be presented in symptoms such as erythematous swelling and pain. Epidemiology reveals bunionette is two to four times more common in women than in men and usually occurs in adolescents and adults. Bilateral feet bunionette deformity is also very common. The etiologies of symptomatic bunionette can be divided into intrinsic and extrinsic factors. Intrinsic factors include anatomic variations such as prominent metatarsal head, lateral bending of the metatarsal shaft, increased four-to-five intermetatarsal angle (IMA), splay foot deformity, congenital plantarflexed or dorsiflexed metatarsal, and soft tissue problems such as keratosis and bursitis [1,2,3]. Extrinsic factors include excess loading on the lateral aspect of the foot and constrictive footwear [4]. The initial management of bunionette is conservative treatment, including changing footwear, semirigid shoe inserts, metatarsal pads, metatarsal bars, and medication such as nonsteroidal oral analgesics and steroid injections [5]. For those in whom conservative treatment fails to relieve the pain, operative intervention should be considered. The goal of operative treatment is to decrease the lateral prominence of the fifth metatarsal. Although effective operative treatments have been described for bunionette deformity, ranging from a simple bunionette resection to complex metatarsal osteotomies, even combined with soft-tissue reconstructive procedures [6,7,8], there is no consensus of operative management. Usually, operative treatments vary due to severity and the deformity leading to lateral prominence. The purpose of this study was to evaluate the radiographic and clinical outcomes of a sliding oblique metatarsal osteotomy (SOMO) for all types of bunionette deformity.

## 2. Methods

We retrospectively reviewed 44 patients (34 females and ten males; 51 feet, 7 bilateral cases, left/right: 29/22) from December 2010 to December 2018 at our hospital. All procedures were performed by two experienced orthopaedic foot and ankle surgeons. The surgical intervention of the bunionette was performed after at least 3 months of conservative treatment failed to treat symptoms. Mean age of patients was 51.01 years old. Exclusion criteria were neuropathy, skin ulcers on lateral side, previous surgeries on lesser toe, and sequelae from fractures of the tibial pilon, ankle, or foot. Most of patients were female, accounting for 77.3% of all cases. Mean follow-up was 26.6 months (minimum 18 months).

Bunionette can be radiographically classified based on Coughlin classification [9] as type I: enlargement of the fifth metatarsal (MT) head or dumbbell-shaped fifth MT head (MT width > 13 mm); type II: lateral bowing of the distal aspect of the fifth metatarsal (lateral deviation angle (LDA) > 7 degrees); type III: increased intermetatarsal angle (IMA) between the fourth and fifth metatarsal (IMA > 8 degrees); Patients who cannot be classified into any of the traditional categories can be placed into the type IV, which was proposed by Fallat et al. [10]. Type IV is the combination of type I with type II or type III.

### 2.1. Outcome Measures

Outcome evaluation in this study was radiographically and clinically performed. In the radiographic outcome, 4–5 intermetatarsal angle (IMA), as well as lateral deviation angle (LDA) and metatarsal phalangeal angle (MTPA), were preoperatively measured and measured at the last postoperative visit to the outpatient clinic using the anteroposterior radiographic images of the weightbearing foot. IMA is the angle formed by two lines that bisect the fourth and fifth metatarsals. Normal value should be less than 8 degrees. LDA is the angle formed by the line from the center of the metatarsal head and the neck to the metatarsal base and the line along the medial cortex of the fifth metatarsal shaft. Normal LDA ranges from 0 to 7 degrees. MTPA is measured by the degree of divergence between long axis of the fifth proximal phalanx from the long axis of the metatarsal shaft, and the value is usually less than 14 degrees in asymptomatic patients. Two doctors, the same as those who conducted the classification, evaluated images and measured these parameters.

In the clinical outcome evaluation, a 100-point The American Orthopedic Foot and Ankle Society (AOFAS) score for lesser metatarsophalangeal procedures was used for subjective and functional outcomes. AOFAS score can be divided into three main categories—pain, function, and alignment—to represent clinical satisfaction after treatment. Visual analog scale (VAS) pain score was also used to reflect the pain improvement, which is the main symptom and discomfort of bunionette patients.

Nonunion was defined as no bone healing in 3 months without any radiographic evidence. Nonunion was evaluated by the last postoperative image of the weightbearing foot anteroposterior radiograph at the outpatient clinic by two previous surgeons.

### 2.2. Operative Technique and Postoperative Care

Patients were set under general anesthesia in the supine position with the application of a tourniquet. Skin of the lower limb on the operated side was sterilized and draped. Initially, a small longitudinal skin (1.5 cm~2 cm) incision was made along the dorsal aspect of fifth metatarsal neck. After deep dissection with caution of extensor digitorum tendon, the periosteum was also incised and reflected. The next step was to make a mean 45 degrees oblique osteotomy from distal–lateral to proximal–medial direction by standard pneumatic mini-saw from the dorsal to plantar aspect. The distal fragment was shifted medially about half of the width of 5th metatarsal shaft. Additionally, the distal fragment was shifted slightly dorsally but remained in contact with the ventral cortex of 5th metatarsal shaft. After sliding, the distal fragment was then fixed by a 1.5 mm Kirschner wire (K-wire). The 1.5 mm K-wire was first antegradely reamed through the medullary canal of the distal fragment and came out from the tip of 5th distal phalanx. We retracked the K-wire from the distal end and kept enough length to perform subsequent fixation, and then we cut the proximal part of the K-wire with a sharp bite. The cut K-wire was retrogradely reamed into the proximal part of 5th metatarsal shaft through metatarsophalangeal joint after sliding medialization (Figure 1). Then, the alignment of osteotomy and the position of 1.5 mm K-wire were checked under fluoroscopy. After normal saline irrigation to clean surgical area, wound closure by mattress sutures was performed with Nylon 4-0 (Unik, New Taipei City, Taiwan). The distal end of the K-wire was cut and bent to prevent skin irritation. Neither short leg splinting and casting nor orthosis were postoperatively used.

After operative procedure, partial weightbearing (less than 50% body weight supported by affected foot) was permitted. The patient received a follow-up at the outpatient clinic at the 1st week, 2nd week, 6th week post operation. K-wire removal was performed 6 weeks later at the outpatient clinic without any anesthesia. After removal, we asked patients to follow up at an interval of 2 weeks. Union was confirmed both clinically as a pain-free osteotomy site and radiographically by the weightbearing foot anteroposterior radiographs. Further clinical visits were arranged in 1~2 months intervals thereafter (Figure 2).

### 2.3. Statistical Data Analysis

The categorical data (data of all patients and subgroup data of each type) are reported as mean, standard deviation and mean change after operation. The Shapiro test was performed to determine whether the data were parametric or nonparametric, and then the means were compared with the nonparametric Mann–Whitney U test. Paired t-test was used for the comparison of the preoperative and postoperative radiographic angles as well as clinical scores. Statistical analysis was carried out with Microsoft Excel 365 (Microsoft, USA). The *p* value < 0.05 was considered statistically significant.

## 3. Results

### 3.1. Patient Demographics

All 44 patients (51 feet) underwent sliding oblique metatarsal osteotomy (SOMO) and received clinical follow-up. After classification by two doctors who were not involved in the operation through radiographic evaluation, there were six type I, ten type II, twelve type III, and twenty-three type IV cases. Most cases of the fifth metatarsal deformities were classified as type IV, representing 45.1% of the sample. Basic information for all cases is shown in Table 1. Mean follow-up period was 26.6 months. Minimal follow-up period was 18 months.

### 3.2. Radiographic Assessment

The mean value of variables, including IMA, LDA, and MTPA were preoperatively and postoperatively compared by a paired t-test and are shown in Table 2. The preoperative (pre-OP) and postoperative (post-OP) mean value differences between IMA, LDA, and MTPA were 3.96, 3.01, and 9.23 degrees. All radiographic measurements achieved significant difference (*p* < 0.001).

### 3.3. Clinical Assessment

In this study, the clinical outcomes, namely the AOFAS and VAS scores, were compared before and after the operative procedure with an average value using a paired t-test. The data are presented in Table 3. The mean values of preoperative AOFAS and VAS scores were 65.43 and 4.14. The mean values of postoperative AOFAS and VAS score were 93.0 and 0.67. 

The average AOFAS score improved with an increase of 27.57 after the SOMO procedure. The average VAS score also showed a reduction of 3.47 after the SOMO procedure. The clinical outcome revealed a significant (*p* < 0.001) improvement after the operation.

### 3.4. Subgroup Analysis

All cases were separated into four groups according to Coughlin and Fallat classification. Then, clinical and radiographic outcomes of subgroupstrac were preoperatively and postoperatively compared again. The data are presented in Table 4. As shown, each outcome assessment reached a significant difference (*p* < 0.05) after operative procedure excluding LDA of type I group (*p* > 0.05).

### 3.5. Complication

There was no nonunion noted in our case series. However, three cases underwent pin-tract wound infection before K-wire removal (infection was noted at mean 1.34 weeks postoperatively). After removal of K-wire and oral antibiotic use for 7 days, all three cases recovered with good wound healing. In addition, no further septic nonunion or osteomyelitis was noted. During the follow-up period, all cases were clinically stable with bone union. No further wound healing problems were found in our cases after K-wire removal. No recurrence of deformity was noted during our follow-up. Though the SOMO may lead to a mild shortening of the fifth ray, no patient experienced fourth metatarsalgia in this study. Patients could wear their regular shoes after removal of stitches and wound healing.

## 4. Discussion

Bunionette is a well-known disease in orthopedics. However, the operative treatments of bunionette remain debated and vary from a simple bunionette resection to complicated metatarsal osteotomies [11,12,13,14,15,16,17,18,19].

Kitaoka et al. provided a simple lateral eminence resection, also called a condylar bunionectomy, as treatment to those patients who had isolated metatarsal head prominence. Condylar resection does not correct malalignment but rather reduces its main effect. Therefore, bunionectomy is usually used for the patients with prominent fifth metatarsal head or non-candidate for osteotomy. A shorter recovery period was found in patients receiving bunionectomy. Moreover, joint mobility and metatarsal length were preserved without osteotomy, avoiding complications associated with osteotomy such as malunion and nonunion. Case series also presented a good satisfaction (71%) after condylar resection with no notable correction of angular deformity and no relationship between the amount of resection and patient satisfaction [18].

Metatarsal osteotomies and bony realignment have been described at the distal, diaphyseal, and proximal levels. Usually, levels of metatarsal osteotomies were selected according to the types and grade of bunionette deformity. Distal metatarsal osteotomies have evolved over time and have more corrective power than simple lateral condyle resection alone [11,16,20,21,22,23,24]. Although transverse and oblique distal osteotomies are attributed to an increased risk of instability, malunion, and a high potential of recurrence, Cooper and Coughlin [25] reported that, in a study of 14 patients with type I deformity treated with subcapital oblique osteotomy, the rate of good or excellent clinical results was 88% at a mean follow-up of 2.9 years. Further reliable distal osteotomies were designed and studied such as distal chevron osteotomy [21]. Distal osteotomy increases the risk of injury of metatarsal head blood supply, which means a higher risk of avascular osteonecrosis [19].

In patients with an increased IMA or substantial lateral bowing of the metatarsal shaft, which means type II or type III deformity, diaphyseal fifth metatarsal osteotomy is indicated [26]. Generally, complication like malrotation, nonunion, and malunion, have been associated with transverse osteotomies. Modified oblique osteotomies were well studied by many authors, and good results were presented in their articles [6,15,27,28].

Proximal osteotomies were considered to offer the greater potential for angular correction but also lead to an increased risk of instability and metatarsal length shortening, which means a higher risk for malunion, nonunion and transfer metatarsalgia. Proximal osteotomy was usually applied to patients with increased IMA, which was a feature of type III or IV bunionette deformity. Okuda et al. [29] reported on a series of 10 patients who underwent proximal third osteotomy of the fifth metatarsal for bunionette correction with sustained correction of a large IMA (12.2° preoperatively to 4.8∘ postoperatively).

The SOMO procedure follows the principle of a minimally invasive technique to optimally reduce the damage to soft tissue and create a sufficient surgical field for oblique osteotomy. Some similar minimally invasive techniques were widely proposed in recent research. These procedures varied from osteotomies and fixation device.

S.E.R.I. (simple, effective, rapid, inexpensive) techniques was first proposed for hallux valgus correction with satisfactory results [30] and has been successively applied to bunionette as a surgical treatment [31]. The S.E.R.I. technique is to perform an oblique osteotomy on fifth metatarsal neck and then apply a 2 mm K-wire into the proximal metatarsal canal as a lever to medially translate the metatarsal head. There is no fixation applied on the cut distal fragment, which is the main difference between the SOMO procedure and S.E.R.I. technique. We believe that the application of a K-wire can provide better stability. Moreover, the angle of inclination when performing oblique osteotomy is also different in that the inclination in the SOMO procedure is larger than the S.E.R.I. technique.

Minimally invasive DMMO (distal metatarsal metaphyseal osteotomy) has proved to be an alternative surgical approach to reduce deformity [32]. DMMO is a kind of percutaneous extraarticular metatarsal neck osteotomy without any internal fixation. The oblique osteotomy of DMMO is performed from the distal–dorsal to proximal–plantar direction, which permits the metatarsal lengths to be automatically set upon weight bearing. The design of the inclination of oblique osteotomy and the concept of fixation are also different between the SOMO and DMMO procedures.

We chose oblique osteotomy in the SOMO procedure because the slope created by oblique osteotomy can guarantee the expected medialization of the distal fragment with little chance of lateral subluxation, preventing the spring-back effect during the remaining steps of the SOMO procedure. In our experience, we applied SOMO to all types of bunionette deformity. The level of osteotomy we selected in the SOMO procedure is between subcapital and diaphyseal osteotomy. The SOMO procedure has a corrective power and effect on the same benefit with a subcapital and diaphyseal osteotomy. Diaphyseal metatarsal osteotomy is indicated for type II or type III, and distal osteotomy is indicated as correct in type I or type II deformity [2]. We believe that this is the possible reason why the SOMO procedure is suitable for all types of deformity.

In our study, we evaluated our cohort by radiographic and clinical assessment, which are extensively used in other studies. IMA, LDA, MTPA can be regarded as deformity degree of bunionette, and symptom improvement is assessed by AOFAS and VAS score. All assessment parameters showed significant improvement (*p* < 0.001) after operative procedure under mean follow-up periods of 26.6 months. SOMO, which is a relatively simple procedure, is useful for providing a deformity correction and satisfactory functional improvement. Moreover, we introduced a subgroup study according to Coughlin and Fallat types, and all assessment parameters received significant (*p* < 0.05) improvement except LDA of type I subgroup. The results of the subgroup analysis showed that the SOMO procedure was effective for different types of deformity. In terms of LDA, there was no clear lateral bowing of the diaphyseal metatarsal shaft in type I, in which deformity is mainly an enlargement of the metatarsal head. Therefore, SOMO seemed to provide no obvious correction with type I in our cohort.

There are several advantages of the SOMO procedure. One advantage is a shorter protective and recovery time in our postoperative protocol. In other articles, most authors suggest a partial weightbearing with a postoperative supporting shoe or orthosis for at least 2 weeks, even for a non-weight-bearing or on-heel gait [6,15,25,27,28]. Secondly, SOMO procedure only requires a small incision for osteotomy, and this procedure is easily reproducible. This means a shorter operation time and easy wound care, which reduces the cost of hospitalization. Additionally, if patients have a problem with the fifth plantar metatarsalgia, we can perform the additional step of slightly dorsal sliding after osteotomy with a dorsal wedge osteoectomy to align the dorsal cortex of the proximal part. Plantar metatarsalgia can also be solved at the same time. The other advantage is that the fixation device in our study is a 1.5 mm K-wire, which is an accessible device that is easy to remove at the outpatient clinic without any anesthesia and hospitalization.

One of the limitations of our procedure is that the degree of medialization was experience-dependent. However, inexperienced surgeons can repeatedly check the alignment under fluoroscopy before fixation or even with temporary fixation before definite fixation. The other limitation is the exposed part of K-wire at the tip of fifth toe. The pin-tract infection is a common complication when fixation with a K-wire is used. In our series, there were three cases with pin-tract infection, but all of them recovered without further healing conditions after the removal of the K-wire and prescription of oral antibiotics. The last limitation of our procedure is that we applied the intramedullary K-wire from tip of the distal phalanx to the tarsometatarsal joint. The joint cartilage of distal interphalangeal, proximal interphalangeal joint, metatarsophalangeal joint and tarsometatarsal joint were injured during the SOMO procedure. Therefore, there is an increased risk of degenerative arthritis in the fifth toe.

In terms of the study method, some advantages of our study can be presented. Firstly, all cases were selected in the same medical center and operated on by the same two experienced surgeons. Secondly, our case numbers were also relatively larger than other case series regarding operative procedures to treat bunionette.

Some limitations of our study are its retrospective nature and relatively lower case numbers in each subgroup. Small case numbers in subgroups of different types might influence the accuracy of statistical analysis. However, most results of the subgroup analysis seemed to have statistically significant improvements. Moreover, pre-operative classification and perioperative parameter measurements were carried out by the same two doctors. Though the doctors who conducted the evaluation were not involved in the surgery, the same doctors to carry out the image evaluation may have caused some bias in our study. The application of a lower AOFAS toe score as a clinical assessment in our study might be another limitation. Though the AOFAS score is widely used in many articles on bunionette and associated techniques, the validity of the AOFAS score for lesser toe is still controversial.

## 5. Conclusions

The SOMO procedure can be considered as a reliable and simple treatment without severe complications. The SOMO performance can be performed in almost all types of bunionette deformity with satisfactory outcomes, but further studies with more cases are required to clarify the efficacy of the SOMO procedure for each type of bunionette.

## Figures and Tables

**Figure 1 jcm-11-03962-f001:**
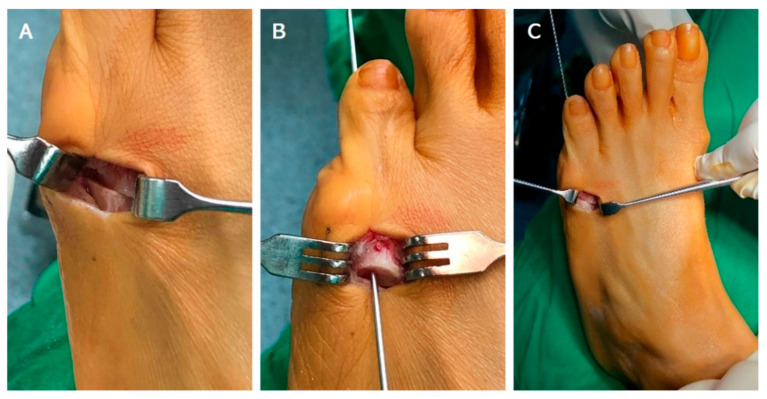
(**A**) Sliding oblique osteotomy was performed on level of 5th metatarsal neck with proper medialization. (**B**) Antegradely reamed 1.5 mm Kirschner through medullary canal to the tip of 5th toe and pull out from the tip for enough length further fixation. (**C**) Fixation was done with retrogradely reamed 1.5 mm Kirschner wire after sliding medialization.

**Figure 2 jcm-11-03962-f002:**
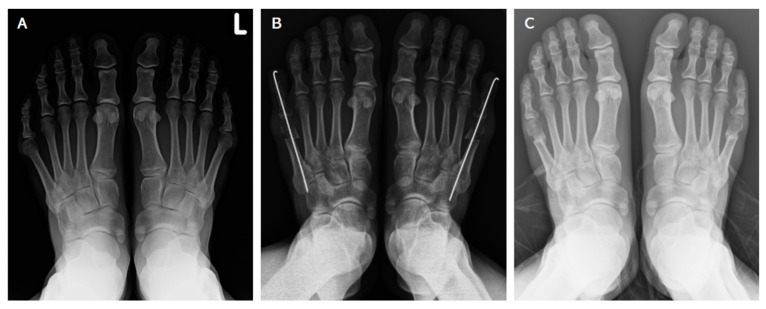
(**A**) Preoperative weightbearing anteroposterior radiographic image of both feet. (**B**) Anteroposterior radiographic image of both feet obtained immediately after the procedure. (**C**) Weightbearing anteroposterior radiographic image obtained 6 months after the procedure, showing bone union with good healing.

**Table 1 jcm-11-03962-t001:** Patient demographic.

Cases	Side(L/R)	Gender	Mean Age	Coughlin and Fallat Type
*n*: 51	29/22	F = 34, M = 10 (7 bilateral feet)	51.01	Ⅰ = 6 Ⅱ = 10 Ⅲ = 12 Ⅳ = 23

Abbreviations: L, left; R, right; F, female; M, male.

**Table 2 jcm-11-03962-t002:** Results of the pre- and postoperative radiographic measurement.

Radiographic Measurements	Preoperative Mean (SD)	Postoperative Mean (SD)	Preoperative and Postoperative Difference	*p* Value
IMA	9.84 (2.40)	5.88 (2.29)	3.96	*p* < 0.001
LDA	4.99 (1.85)	1.98 (1.41)	3.01	*p* < 0.001
MTPA	15.72 (5.50)	6.49 (4.77)	9.23	*p* < 0.001

Abbreviations: IMA, 4th to 5th intermetatarsal angle; LDA, lateral deviation angle; MTPA, metatarsal phalangeal angle.

**Table 3 jcm-11-03962-t003:** Results of the pre- and postoperative clinical scores.

Clinical Scores	Preoperative Mean (SD)	Postoperative Mean (SD)	Preoperative and Postoperative Difference	*p* Value
AOFAS	65.43 (10.04)	93 (10.35)	27.57	*p* < 0.001
VAS	3.83 (1.02)	0.17 (0.91)	3.66	*p* < 0.001

Abbreviations: AOFAS, The American Orthopedic Foot and Ankle Society for lesser metatarsophalangeal procedures; VAS, visual analog scale.

**Table 4 jcm-11-03962-t004:** Subgroup results of the pre- and postoperative radiographic measurement and clinical scores.

Subgroup Type	IMA Mean Difference	LDA Mean Difference	MTPA Mean Difference	AOFAS Mean Difference	VAS Mean Difference
Type I *n* = 6	2.91	2.18	11.33	29.66	3.66
*p* value	*p* < 0.001	*p* > 0.05	*p* < 0.001	*p* < 0.001	*p* < 0.001
Type II *n* = 10	3.31	2.34	8.33	26.0	3.40
*p* value	*p* < 0.001	*p* < 0.05	*p* < 0.001	*p* < 0.001	*p* < 0.001
Type III *n* = 12	4.15	2.80	7.04	25.83	3.08
*p* value	*p* < 0.001	*p* < 0.001	*p* < 0.001	*p* < 0.001	*p* < 0.001
Type IV *n* = 10	4.37	3.61	10.21	28.31	3.61
*p* value	*p* < 0.001	*p* < 0.001	*p* < 0.001	*p* < 0.001	*p* < 0.001

Abbreviations: IMA, 4th to 5th intermetatarsal angle; LDA, lateral deviation angle; MTPA, metatarsal phalangeal angle; AOFAS, The American Orthopedic Foot and Ankle Society for lesser metatarsophalangeal procedures; VAS, visual analog scale.

## Data Availability

The datasets used and/or analyzed during the current study are available from the corresponding author on reasonable request: Dr. Peng-Ju Huang (roger01@ms4.hinet.net; Tel.: +886-939-347960).

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
