# Peer review of "Symptomatic Bunionette Treated with Sliding Oblique Metatarsal Osteotomy—Case Series"

_jcm, 2022, doi:10.3390/jcm11143962_

Round 1
Reviewer 1 Report
Abstract-succinct and well written
Intro-well written. adequate citation of relevant literature for background. No hypothesis stated but does not apply to this particular type of study.
Methods-clearly presented. Inclusion and exclusion criteria are clearly stated. Statistical methods appropriate. Line 66-69 belong in results section? Good description of outcome measures. The authors should explain why the AOFAS score was chosen as this is not a validated outcome measure.
Operative technique presented well.
Results are presented clearly.
Discussion well written.
Author Response
Please see the attachment, thanks

Reviewer 2 Report
First of all, congratulations on the successful operation of several cases. And you did a good job organizing a lot of stuff.
The overall design of the article is well thought out.
1) However, there are several limitations of this case series. As explained in the last limitation, it seems unreasonable to draw a conclusion because there are few cases of subgruop. Make it clear that future studies with more cases are needed.
2) Next, this operation is considered a type of S.E.R.I operation. It is almost similar to the boesch technique, but what makes this operation different from other similar osteotomy?
3) What are the reasons or advantages of the authors doing oblique osteotomy compared to transverse osteotomy? Please explain this clearly to your readers.
4) The important point is to explain and understand the differences and advantages of the authors' technique compared to other surgical methods to the reader.
This has a different meaning from radiologic or clinical results. Because similar surgical techniques and results have already been reported, it is necessary to emphasize the strengths and differences of the authors' own techniques.
